# The patient safety curriculum: An interventional study on the effectiveness of patient safety education for Jordanian nursing students

**Ahmad H. Al-Nawafleh**[1]* , **Sultan Musleh**[2] , **Nofal Nawafleh**[3]

1 Associate Professor of Nursing Management and Leadership, Faculty of Nursing, Mutah University, Mu'tah, Jordan, 2 Associate Professor of Adult Health Nursing, Faculty of Nursing, Mutah University, Mu'tah, Jordan, 3 Faculty of Medicine, Mutah University, Mu'tah, Jordan

☯ These authors contributed equally to this work.
* alnawafleh@mutah.edu.jo

## Abstract

### Introduction

The purpose of the study was to assess the effectiveness of the World Health Organization's (WHO) patient safety curriculum (PSC) in improving the patient safety education for nursing students in Jordanian higher education programmes.

### Methods

An interventional design adopting a pre-test and post-test was used. A questionnaire survey was administered to 373 nursing students before and after the curriculum. Students were asked to complete pre-test and post-test questionnaires on self-awareness of patient safety.

### Results

Students had moderate knowledge of factors that influence patient safety and scored as the highest, (mean 3.45, SD 0.94). The greatest improvement was in the role of the health organisation in error reporting (the mean difference was 0.97, P = 0.001). Taking into consideration the essential measurements, the results obtained after the post-test regarding the two patient safety topics showed a significant increase after completing the training, demonstrating that the patient safety course was effective.

### Conclusion

The study highlights the importance of developing a curriculum in nursing schools that incorporates patient safety education. The WHO PSC guide can be a great start in this domain.

**Data Availability Statement:** All relevant data are within the paper and its Supporting Information files.

**Funding:** This study was partially funded by Mutah University (https://www.mutah.edu.jo) with the gr [120/14/114/ ع.ب]. The fund was received by Ahmad Al-Nawafleh. The funder had no role in the study design, data collection and analysis, decision to publish, or preparation of the manuscript.

**Competing interests:** The authors have declared that no competing interests exist.

# Introduction

Patient safety and nursing mistakes are now a global concern for providing high quality health care. In 1999, the Institute of Medicine (IOM) released a starting report on the number of deaths caused by medical errors annually in the United States [1]. According to the report titled "*To Err Is Human*: *Building a Safer Health System*", by the IOM, It was approximately that medical errors caused around 98,000 fatalities on a yearly basis [1–3]. The report warns policy makers, hospitals, health care staff and patients about the main problems related to patient safety. The global focus on mitigating the negative impacts of healthcare is of utmost importance, and in the last ten years, there has been significant increase in patient safety efforts due to advances in patient safety sciences [4–8]. While, it is impossible to predict when a patient harm may occur, there are measures that can be considered to prevent it from happening again. Healthcare professionals and organizations should learn from previous mistakes to avoid future commitments of them.

However, the provision of healthcare in certain situations continues to be potentially dangerous despite earlier efforts [9, 10]. Patient safety experts conclude that a major reform of nursing education in nursing schools and resident training programmes is essential for essential improvements in patient safety [11–13].

Despite the significance of ensuring patient safety and the necessity of raising awareness about it, only a small number of educational institutions include this subject in their syllabus [6, 11, 14]. Historically, the educational program for nursing and medical students has concentrated exclusively on practical abilities; such as identifying illness, providing treatments, providing post treatment care and monitoring [9, 12, 15, 16]. However, the concepts of communication skills, root cause analysis, system thinking, human factor science has been disregarded to a great extent [14, 17, 18]. It is essential for patient safety that all nursing students at the undergraduate level to possess all these essential skills and the required proficiency to reduce any harm to patients [3, 10]. It is urgent to integrate knowledge and skills about patient safety in the health care field into the nursing student curriculum.

Patient safety is becoming a standard that is common in global and local healthcare practice. However, many nurses still do not adequately practice it. Furthermore, it is not widely taught to nursing students [16]. The WHO developed a curriculum to help the nurses of the future to practice PS and thus help create a culture of safety for all patients [4].

It is important when adopting a training curriculum to find out whether the students can achieve the competencies required after its application. Furthermore, it is important to find out how the training will fit into the curriculum of the nursing program. As a result, nurses' programme developers should have a list of core training objectives based on agreed core competencies and examples of training programs selected by teachers on the basis of the core training objectives and target groups. When designing a training program, the developer must decide which of these skills should be taught. Selecting a level depends on the expected level of competence and characteristics of the student (e.g. previous training in PSC, research methods, epidemiology and statistics), as well as the level of training necessary to establish basic knowledge and understanding of nursing.

In Jordan, nursing education has been identified as a priority and efforts have been made to improve the quality of nursing education. The PSC was introduced into undergraduate nursing programs in Jordan as a strategy to improve patient safety education for nursing students [19]. This study aimed to examine the impact of PSC on nursing students' knowledge, skills, and attitudes towards patient safety. This curriculum design is based on the patient safety guide proposed by the World Health Organization [4].

There is no formal educational training specifically for patient safety in Jordanian universities' nursing programs up to the time of writing this report. However, the topic of patient safety is mentioned sporadically in the nursing curriculum, often related to ensuring that the student nurses are being competent in their practice [19–21].

There is now a new drive to reform the nursing curriculum in our institution to ensure that we incorporate the most relevant teaching topics but also use the most appropriate teaching methods. Patient safety, along with other topics such as palliative care and health information systems, is on the top list of topics that we need to integrate into our curriculum. Therefore, "the Multi-professional Patient Safety Curriculum Guide" [4] was a good opportunity to start working in this direction.

The opportunity to participate in this worldwide assessment as a "Complementary Test Site" is open to all schools and universities that specialize in nursing, dentistry, medicine, midwifery, pharmacy, and other healthcare professions [4]. These institutions should be interested in incorporating topics from this guide that were adopted in similar studies [15, 17, 22, 23]. These universities and schools should be recognized by the national regulatory bodies of their country and be able to demonstrate an interest in teaching patient safety [4].

The university under study is now reforming its curriculum for preregistration nursing training according to the University Deans Council recommendations. The new curriculum includes new topics to be taught to prospective students, such as palliative care, end-of-life care and patient safety. Therefore, the current guide provides an important tool for the university to utilize in integrating patient safety education into our pre-registration nursing curriculum.

This study aims to evaluate the effectiveness of teaching the WHO PSC to undergraduate nursing students in the Jordanian context. Therefore, it has two objectives; to determine:

- The value of the curriculum guide for students. The usability of the curriculum guide as an educational tool for health education in universities and schools in Jordan.

- Impact/effect of the curriculum guide on student knowledge of patient safety in Jordan.

## Literature review

The WHO PSC Guide: "the multi-professional edition" provides a common platform for nurses to address their role to play in improving patient safety [4]. Although nursing professionals must ensure the practice does not harm, evidence indicates that nursing education is still lagging to integrate patient safety into their curriculum [16]. Therefore, this guide highlights understanding of the need to integrate the safety of patients into undergraduate nursing curriculum.

Huang and her colleagues [24] explored the factors that affect the nursing students competence of PS who completed a national online survey. They recruited 732 Chinese students during 2018 and 2019 across various locations [24]. The main outcome factor was the rating of the patient safety survey. The study found that Chinese nursing graduates showed moderate confidence in their clinical safety skills, but lacked confidence in areas such as learning about socio cultural or context dependent patient safety (PS) and discussing PS related topics like effective communication and understanding human and environmental factors [24]. Additionally, less than half of the participants felt comfortable talking to individuals who engaged in unsafe practices and expressing concerns about adverse reactions. Overall, these findings shed a light on the PS skills of Chinese nursing graduates.

In a study conducted by Suliman [25], the level of PS literacy among nursing students in two Jordanian universities was assessed. The study found that PS issues are not given much

importance in the nursing education and curricula in Jordan. The study was conducted through a cross-sectional survey involving 297 nursing students. The results indicated that the students had moderate satisfaction with their knowledge and skills on most parameters. They were more confident in the knowledge and skills learned in the classroom compared to clinical training. Additionally, senior students lacked confidence in their knowledge and skills, particularly when it comes to writing medical error reports. The study suggests that contextual and sociocultural issues related to PS should be emphasized in the nursing curriculum to enhance students' competence in speaking up when they witness healthcare errors [25]. To bridge the gap between theory and practice, the study recommends adding PS elements to the nursing curriculum.

A study was conducted in Jordan to measure the perceptions of PS during four consecutive semesters [19]. It emphasised the inclusion of PS in training programmes and nursing education. PS in the nursing curriculum will emphasise the importance of reducing medical errors, highlighting clinical excellence, and improving health outcomes. The study was a survey of 158 nursing students in their final year. It found that only 25% of the participants had used an incident report for the errors they observed. However, during their practice in clinical practice, 62% of students.

## Methods

Interventional study adopting the pre-test and post-test method.

### Study setting and sample

The participants who took part in the investigation were nursing students from a public university located in Jordan. Those contacted during the first semester of 2019–2020. A total of 373 students attended this PS program and their perception changes to the PS program were surveyed before and after the programme.

### The intervention

The WHO developed training and we have chosen two topics from 11 topics because of limited funding and resources. The intervention was based on the WHO PSC guide for multidisciplinary health professions [4, 13, 16, 17]. We taught selected topics from this curriculum to all nursing students as part of several courses during their studies. The methodology involved administering pre-test and post-test questionnaires to students to assess their self-awareness of PS before and after the curriculum. We conducted training on topic 1: what is patient safety? and topic 9: Infection prevention and control (World Health Organization 20011) [16]. The lectures were originally designed by the WHO team, but we adapted them to our Jordanian situation using contemporary health incidents as illustrative cases. These incidents were handled individually by our staff, who were able to provide detailed analysis and real management strategies.

### The study tool

The survey used in the research was adapted and modified with permission from the WHO and the investigation was conducted adhering to their established protocols [4]. The tool validity and reliability tested by several studies and assured through multiple locations evaluations which supported by the WHO [16, 26, 27]. It consisted of 23 items that were organized into four groups or composites. The first composite consisted of seven items and was focused on assessing the student's knowledge of PS. The second composite assessed the Safety of the

Healthcare System and was made up of five items. The third composite that evaluated the student's ability to influence PS contained seven items. Finally, Composite 4 which was composed of four items, focused on the students' Personal Attitudes to PS. To rate the students' responses, a 5-point ordinal scale was used, with "1" indicating "strongly disagree/very poor" and "5" indicating "strongly agree/very good" [4].

Knowledge questions were used to determine what students have learnt about PS. Two to three knowledge questions have been used for each of the topics of the curriculum guide. Students have been asked the knowledge questions at the start of the courses on PS topics to establish baseline data on students' knowledge. Their responses to the knowledge questions at the end of the study were compared with baseline data, to quantify changes in student knowledge as a result of their learning certain PS topics.

## Study ethics and actions

The study was approved by the university, the Scientific Research Ethics Committee, with reference number (SREC-7-2019). A voluntary self-guided survey was conducted among 373 nursing students. The survey was conducted during September 2019 and three months later during December 2019. Both were given during the entire class in the lecture room. Students have about 10–25 minutes to complete the questionnaire. The consent to take part in the study was voluntary and granted before the student responded to the questionnaire based on information given orally during the class.

## Analysis

SPSS ver.25 was used to analyse the data. Descriptive statistics and paired t-tests were used in data analysis.

## Results

The study invited 373 students to participate, and 253 (68%) questionnaires were returned for the pre-test. Two months after the last learning session or three months after the first session, 218 (58%) completed and returned the post-test questionnaire (see Fig 1). Only forty students (16%) reported having prior exposure to PS training, while 172 (68%) acknowledged that their university had taught PS content (see Table 1).

### Students' attitudes before patient safety training

Before the training courses took place, Table 2 illustrates the knowledge, skills, and attitudes (KSA) that the students had about PS. The feedback from the students indicated low level of knowledge in the first composite regarding PS except for factors that contribute to human error or influence PS, where they possessed moderate knowledge. The second composite is the Safety of the Healthcare System evaluated by five items. Most of the students reported a neutral position although some agreed and others disagree. In response to the third composite, the students agree that they can influence PS. The items measure their influence, which may indicate attitude rather than knowledge. This is confirmed by the student's responses to the last composite, as all students indicated that they strongly agree on personal attitudes toward PS.

### Student attitudes after patient safety training

In Table 2 the results of the PS perception survey are presented. Students who had a moderate understanding of the factors that affect PS achieved the highest scores (mean 3.45, SD 0.94). In contrast, the lowest score was related to the role of health institutions in error reporting (mean

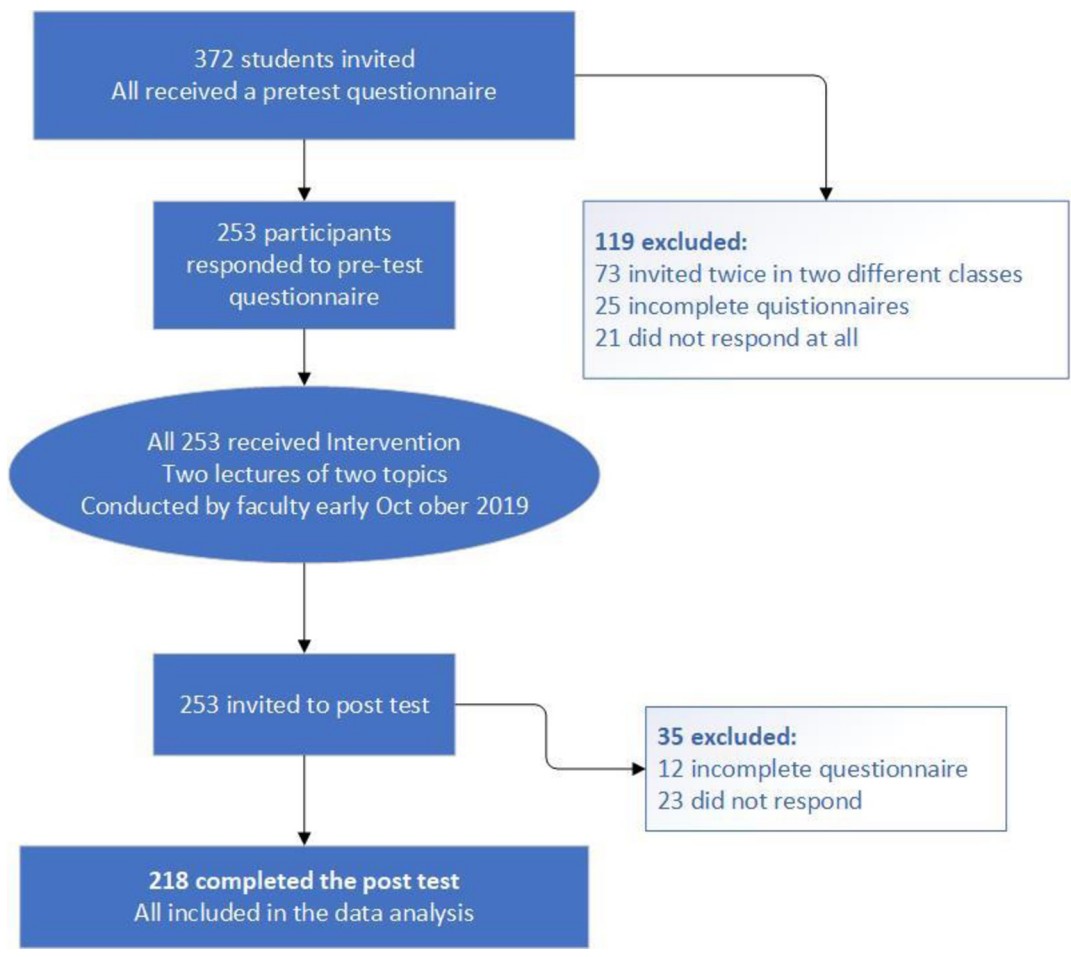

**Fig 1. Study process flow chart.**

1.90, SD = 0.97). This indicates insufficient knowledge among students about the role of healthcare institutions in reporting errors. Additionally, the students received a low score in reporting error. Following the completion of the training, the students' knowledge about patient safety improved significantly, and most items receiving scores greater than three, indicating good knowledge. After the training, five out of seven items showed significant

**Table 1. Demographic data.**

| | | Frequency | Percentage % |
|---|---|---|---|
| **Study level** | 2nd year | 92 | 36.2 |
| | 3rd year | 65 | 25.6 |
| | 4th year | 93 | 36.6 |
| **Has patient safety been taught in your university /school?** | Yes | 172 | 67.7 |
| | No | 80 | 31.5 |
| **Have you previously taken a patient safety course?** | Yes | 40 | 15.7 |
| | No | 206 | 81.1 |

Source: study data

**Table 2. Composite 1- Knowledge of patient safety.**

| Survey Item | Pre-Training | | Post-Training | | Correlation | p value |
|---|---|---|---|---|---|---|
| | Mean | SD | Mean | SD | | |
| 1. Different types of human error in health care | 2.92 | .917 | 3.27 | .872 | .034 | .619 |
| 2. Factors contributing to human error | 2.98 | .925 | 3.43 | .854 | .175 | .010 |
| 3. Factors influencing patient safety | 3.45 | .940 | 3.65 | .801 | .129 | .057 |
| 4. Ways of speaking up about error | 2.67 | 1.081 | 3.18 | .877 | .182 | .008 |
| 5. What should happen if an error is made | 2.78 | 1.012 | 3.26 | 1.053 | .151 | .027 |
| 6. How to report an error | 2.34 | 1.144 | 2.92 | 1.105 | .263 | .000 |
| 7. The role of health-care organizations in error reporting | 1.90 | .974 | 2.87 | 1.129 | .241 | .000 |

Source: study data

improvements, with the most improvement observed in the role of healthcare organizations in error reporting (the mean difference was 0.97, $P = 0.001$).

Comparing the means for students' knowledge of the safety of the health care system before and after teaching the curriculum indicates an increase. For example, healthcare personnel who receive PS training scored 2.94 before and 3.13 after training with a standard deviation of (1.07 and 0.97) consequently. However, this increase is not statistically significant when we asked the students (Item 4) how often the patients received the wrong drug. (see Table 3).

The results of students' influence on safety are not indicating a clear picture. There is a slight increase in the student's ability to talk or tell others about an error they made. This result is statistically significant. Students strongly believed that reporting the error will improve PS, scored as highest (mean = 3.93, SD1.009), students also always ensure that their PS is maintained. After training, the only significant change was in the ability of the students to report their errors (items 1 & 7) (see Table 4).

Personal attitude towards PS did not show a significant change apart from the recognition by students of the importance of learning how best to deal with their errors by the end of nursing school (mean = 4.23 & SD = .8) (see Table 5). All other items were not significant, and they indicate a decrease, too.

## Discussion

The study findings demonstrated that implementing the WHO PSC had a beneficial effect on the attitudes and knowledge of undergraduate nursing students regarding PS. It's pre-test scores revealed that the participants had a positive attitudes and moderate level of knowledge towards PS. However, after the intervention, the post-test scores showed an increase in their

**Table 3. Composite 2- Safety of the health care system.**

| Survey Item | Pre-Training | | Post-Training | | Correlation | p value |
|---|---|---|---|---|---|---|
| | Mean | SD | Mean | SD | | |
| 1. Most health-care workers make errors | 3.33 | .988 | 3.51 | .947 | .257 | .000 |
| 2. In my country there is a safe system of healthcare for patients | 2.94 | .958 | 2.88 | .926 | .236 | .000 |
| 3. Medical error is very common | 3.77 | .945 | 3.74 | .906 | .168 | .013 |
| 4. It is very unusual for patients to be given the wrong drug | 2.77 | 1.070 | 2.91 | 1.118 | .053 | .440 |
| 5. Health-care staff receive training in patient safety | 2.94 | 1.071 | 3.13 | .975 | .211 | .002 |

Source: study data

**Table 4. Composite 3-Personal influence over safety.**

| Survey Item | Pre-Training | | Post-Training | | Correlation | p value |
|---|---|---|---|---|---|---|
| | Mean | SD | Mean | SD | | |
| 1. Telling others about an error I made would be easy. | 3.35 | 1.206 | 3.51 | 1.030 | .189 | .005 |
| 2. It is easier to find someone to blame rather than focus on the causes of error. | 2.07 | 1.104 | 2.24 | 1.074 | .103 | .131 |
| 3. I am confident about speaking to someone who is showing a lack of concern for a patient's safety. | 3.48 | 1.153 | 3.42 | 1.005 | .060 | .377 |
| 4. I know how to talk to people who have made an error. | 3.36 | 1.009 | 3.32 | .951 | .125 | .067 |
| 5. I am always able to ensure that patient safety is not compromised. | 3.89 | 1.040 | 3.85 | .905 | .037 | .591 |
| 6. I believe that filling in reporting forms will help to improve patient safety. | 3.93 | 1.009 | 3.90 | .883 | .080 | .240 |
| 7. I am able to talk about my own errors. | 3.60 | 1.026 | 3.76 | .975 | .252 | .000 |

Source: study data

knowledge and positive attitudes towards PS. These results are consistent with previous studies that have reported the effectiveness of PS education in enhancing healthcare providers' knowledge and attitudes of toward PS [12, 19, 28]. Mansour and his colleagues study results underscore the need for formal PS education and highlight the importance of improving overall safety competence and health outcomes [11].

According to this study, students' ability to identify and talk about their mistakes improved significantly after training. However, their perception towards PS showed a significant change as well, as they recognized the importance of learning how to deal with their errors by the end of their nursing education. This finding is consistent with Suliman [25] research which found that PS is not given enough importance in Jordanian nursing education and curricula. The students had more confidence in their knowledge and skills acquired in the classroom than their practical training in clinical settings, particularly in writing medical error reports [24, 25]. The study concludes that contextual and sociocultural issues related to PS should be emphasized in nursing curriculum to enable students to speak up when they observe healthcare errors. As other studies indicated, to achieve this, it is recommended that PS elements be added to bridge the gap between theory and practice in the nursing curriculum [25].

## Study limitations

Although we obtained a significant difference in the measures before/after teaching the PSC, this is probably not ultimate, the curriculum probably caused the increase in KSA of PS. Other variables may have also influenced this increase [16, 19]. The passage of time between the pre-test and post-test without curriculum teaching could have contributed to this change. Other events may also occur during this period that influenced the students' KSA about PS. Perhaps,

**Table 5. Composite 4- Personal attitudes to patient safety.**

| Survey Item | Pre-Training | | Post-Training | | Correlation | p value |
|---|---|---|---|---|---|---|
| | Mean | SD | Mean | SD | | |
| 1. By concentrating on the causes of incidents I can contribute to patient safety. | 4.13 | .775 | 4.04 | .727 | .130 | .056 |
| 2. If I keep learning from my mistakes, I can prevent incidents. | 4.45 | .665 | 4.24 | .836 | .063 | .353 |
| 3. Acknowledging and dealing with my errors will be an important part of my job. | 4.21 | .775 | 4.15 | .826 | .065 | .339 |
| 4. It is important for me to learn how best to acknowledge and deal with my errors by the end of nursing school. | 4.49 | .687 | 4.23 | .805 | .206 | .002 |

Source: study data

the students were exposed to PS training in clinical settings or informally by the instructors. This means that there is a possibility of contamination from any source that was not under the control of this study [7, 29].

In this study, we used measures to anticipate any confounding factors during the research design. For example, we have asked whether the student has previously been exposed to PS training. However, the results of this study can be improved in future; if researchers use a control group that is not exposed to the intervention and differentiate between the intended intervention and other domains included in the curriculum. In addition, the intervention should be conducted by teachers other than their regular course instructors to avoid using any of the materials during their normal course. This would assist in excluding the effects of time and other factors that could affect the study's results. This study only investigated two domains out of the 11 courses suggested by WHO [4]. Therefore, the results may not be generalised to all other topics of the curriculum suggested by the WHO.

## Implications

A formal PSC adoption and its early integration into the undergraduate nursing curriculum were determined to be necessary according to this study. During the study, we also anticipated that teacher would need to become more involved in the creation and integration of PS in the curriculum. There is a good opportunity to use the "WHO Curriculum Resource Guide" in any attempt to develop an appropriate PSC. The attitude towards PS in academic culture must not only include students, but there has been resistance to implementation at different levels, even from the curriculum regulatory bodies. The WHO curriculum can be amended according to the culture of local institutions and can be properly implemented with the support of the relevant government authorities.

Future research should evaluate the usefulness of WHO's PSC in a wider and more diverse sample group to improve the universality of the results. Finally, it is recommended that more research to be done to determine how the WHO PSC has affected the patient outcomes and how it may reduce adverse reactions in clinical practice.

## Conclusion

In conclusion, this study has shed important light on undergraduate nursing students' attitudes, knowledge and abilities concerning PS, both before and after training. The results indicate that the students had a low level of knowledge in some areas of PS before training, especially with regard to the role of healthcare institutions in reporting errors. However, the training using the WHO PSC significantly improved the students' knowledge and attitudes in most PS areas, with the most significant improvement observed in the role of healthcare organization in error reporting. The curriculum was well received by the participants and they were confident in their ability to apply the knowledge acquired in their future clinical practice. This indicates the need to train nursing students on how to respond to healthcare errors in general, not only to provide safe bedside care to the patients. Overall, the study suggests that training programs can effectively improve students' KSA when included in the curriculum of nursing schools. The WHO PSC guide can be a great start in this domain, which is essential to improve the quality of healthcare services.

## Supporting information

**S1 Checklist. Human participants research checklist.**
(DOCX)

**S1 File.**
(RAR)

## Acknowledgments

The authors thank the University of Mutah, special thanks go to the nursing academics who conducted the teaching and supported the study, and load of thanks to the study participants who completed the questionnaires and who took the time to participate in this study.

## Author Contributions

**Conceptualization:** Ahmad H. Al-Nawafleh.

**Data curation:** Ahmad H. Al-Nawafleh, Sultan Musleh.

**Formal analysis:** Ahmad H. Al-Nawafleh, Sultan Musleh, Nofal Nawafleh.

**Funding acquisition:** Ahmad H. Al-Nawafleh.

**Investigation:** Sultan Musleh, Nofal Nawafleh.

**Methodology:** Ahmad H. Al-Nawafleh, Nofal Nawafleh.

**Project administration:** Ahmad H. Al-Nawafleh.

**Resources:** Ahmad H. Al-Nawafleh.

**Supervision:** Ahmad H. Al-Nawafleh.

**Validation:** Ahmad H. Al-Nawafleh, Sultan Musleh.

**Visualization:** Ahmad H. Al-Nawafleh, Nofal Nawafleh.

**Writing – original draft:** Ahmad H. Al-Nawafleh, Sultan Musleh, Nofal Nawafleh.

**Writing – review & editing:** Ahmad H. Al-Nawafleh.

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
