## [Decision Letter · Decision Letter 0]

29 Feb 2024

PONE-D-23-31102The Patient Safety Curriculum: An Interventional Study on the Effectiveness of Patient Safety Education for Jordanian Nursing StudentsPLOS ONE

Dear Dr. Al-Nawafleh,

Thank you for submitting your manuscript to PLOS ONE. After careful consideration, we feel that it has merit but does not fully meet PLOS ONE’s publication criteria as it currently stands. Therefore, we invite you to submit a revised version of the manuscript that addresses the points raised during the review process.

We look forward to receiving your revised manuscript.

Kind regards,

Omar Mohammad Ali Khraisat, Associate Professor

Academic Editor

PLOS ONE

Journal Requirements:

2. We note you have included a table to which you do not refer in the text of your manuscript. Please ensure that you refer to Table 1 in your text; if accepted, production will need this reference to link the reader to the Table.

Reviewers' comments:

Reviewer's Responses to Questions

**Comments to the Author**

1. Is the manuscript technically sound, and do the data support the conclusions?

Reviewer #1: Partly

Reviewer #2: Yes

2. Has the statistical analysis been performed appropriately and rigorously? 

Reviewer #1: I Don't Know

Reviewer #2: Yes

3. Have the authors made all data underlying the findings in their manuscript fully available?

Reviewer #1: Yes

Reviewer #2: Yes

4. Is the manuscript presented in an intelligible fashion and written in standard English?

Reviewer #1: Yes

Reviewer #2: Yes

5. Review Comments to the Author

Reviewer #1: It would be better to cover most of the aspect of the Multi-professional Patient Safety Curriculum Guide published by WHO in this research.

Better to add more evidence on Literature review related to errors and patient safety.

Rearrange references

Reviewer #2: Thank you so much for this reasonable work and important topic . However, some few changes still required

Firstly, could you emerge introduction, background, and justification of the study in one part to keep the harmony of this work

Secondly, could you provide more information about the intervention

Could you provide more information about the psychometric properties of the instrument that you used

Could you expand the discussion part to include more information about the previous study?

No need for literature part, just used it in the discussion to compare you results with previous studies.

Good luck

6. PLOS authors have the option to publish the peer review history of their article (what does this mean?). If published, this will include your full peer review and any attached files.

Reviewer #1: No

Reviewer #2: No

---

## [Author Response · Author response to Decision Letter 0]

3 Mar 2024

Dear reviewers, 

Thank you for your effort and time. In the following, you will find our reply to each point you raised regarding the manuscript, and we amended it in the manuscript too. This will be identified through track changes file and kept as accepted changes in the file of the “Manuscript”.

Thank you again.

Reviewer #1: It would be better to cover most of the aspect of the Multi-professional Patient Safety Curriculum Guide published by WHO in this research.

We agree with you. However, the constraints which we indicated in the manuscript did not allow for this. We hope that we will secure enough funds in future to support a study of all aspects. 

Better to add more evidence on Literature review related to errors and patient safety.

Rearrange references: Thank you: Done as requested.

Reviewer #2: 

Firstly, could you emerge introduction, background, and justification of the study in one part to keep the harmony of this work

Thank you, we emerged introduction, background, and justification of the study in one part. 

Secondly, could you provide more information about the intervention

The Intervention is added in abstracted manner. However, detailed intervention already exists within the guide published by the WHO in its guide for the evaluation of the curriculum which cited in the text too. 

Could you provide more information about the psychometric properties of the instrument that you used

We indicated this is a tool already prepared by the WHO and tested in several places over the world including some Arab countries. This indication is added to the manuscript in page 9 under the section “study tool” in the first paragraph. 

Could you expand the discussion part to include more information about the previous study?. 

No need for literature part, just used it in the discussion to compare you results with previous studies.

We agree with you that we may delete the literature review section. However, the other reviewer indicated to expand the literature review. Therefore, we would say: more about the previous studies already exist within the literature review section that is why we kept it in the manuscript.

Kind regards

Ahmad

---

## [Decision Letter · Decision Letter 1]

8 Apr 2024

The patient safety curriculum: An interventional study on the effectiveness of patient safety education for Jordanian nursing students

PONE-D-23-31102R1

Dear Dr. ,

We’re pleased to inform you that your manuscript has been judged scientifically suitable for publication and will be formally accepted for publication once it meets all outstanding technical requirements.

Kind regards,

Omar Mohammad Ali Khraisat, Associate Professor

Academic Editor

PLOS ONE

Additional Editor Comments (optional):

Reviewers' comments:

Reviewer's Responses to Questions

**Comments to the Author**

1. If the authors have adequately addressed your comments raised in a previous round of review and you feel that this manuscript is now acceptable for publication, you may indicate that here to bypass the “Comments to the Author” section, enter your conflict of interest statement in the “Confidential to Editor” section, and submit your "Accept" recommendation.

Reviewer #2: All comments have been addressed

2. Is the manuscript technically sound, and do the data support the conclusions?

Reviewer #2: Yes

3. Has the statistical analysis been performed appropriately and rigorously? 

Reviewer #2: Yes

4. Have the authors made all data underlying the findings in their manuscript fully available?

Reviewer #2: Yes

5. Is the manuscript presented in an intelligible fashion and written in standard English?

Reviewer #2: Yes

6. Review Comments to the Author

Reviewer #2: Thank you for your efforts, your paper will add to the literature to this area

No more revisions are required

7. PLOS authors have the option to publish the peer review history of their article (what does this mean?). If published, this will include your full peer review and any attached files.

Reviewer #2: No

---

## [Editor Report · Acceptance letter]

26 Apr 2024

PONE-D-23-31102R1 

PLOS ONE

Dear Dr. Al-Nawafleh, 

I'm pleased to inform you that your manuscript has been deemed suitable for publication in PLOS ONE. Congratulations! Your manuscript is now being handed over to our production team.

Kind regards, 

on behalf of

Dr. Omar Mohammad Ali Khraisat 

Academic Editor

PLOS ONE